# Objective and Perceived Risk in Seismic Vulnerability Assessment at an Urban Scale

Eliana Fischer [1,*], Alessio Emanuele Biondo [2], Annalisa Greco [3], Francesco Martinico [4], Alessandro Pluchino [1,5] and Andrea Rapisarda [1,5,6]

1 Department of Physics and Astronomy "Ettore Majorana", University of Catania,
Via Santa Sofia, 64, 95123 Catania, Italy; alessandro.pluchino@ct.infn.it (A.P.); andrea.rapisarda@unict.it (A.R.)
2 Department of Economics and Business, University of Catania, Corso Italia, 55, 95129 Catania, Italy;
ae.biondo@unict.it
3 Department of Civil Engineering and Architecture, University of Catania,
Via Santa Sofia, 64, 95123 Catania, Italy; annalisa.greco@unict.it
4 Department of Agriculture, Food and Environment, University of Catania,
Via Santa Sofia, 98, 95123 Catania, Italy; francesco.martinico@unict.it
5 INFN Sezione di Catania, Via Santa Sofia 64, 95123 Catania, Italy
6 Complexity Science Hub, 1080 Vienna, Austria
* Correspondence: eliana.fischer@unict.it; Tel.: +39-392-484-9976

**Abstract:** The assessment of seismic risk in urban areas with high seismicity is certainly one of the most important problems that territorial managers have to face. A reliable evaluation of this risk is the basis for the design of both specific seismic improvement interventions and emergency management plans. Inappropriate seismic risk assessments may provide misleading results and induce bad decisions with relevant economic and social impacts. The seismic risk in urban areas is mainly linked to three factors, namely, "hazard", "exposure" and "vulnerability". Hazard measures the potential of an earthquake to produce harm; exposure evaluates the size of the population exposed to harm; and vulnerability represents the proneness of considered buildings to suffer damages in case of an earthquake. Estimates of such factors may not always coincide with the perceived risk of the resident population. The propensity to implement structural seismic improvement interventions aimed at reducing the vulnerability of buildings depends significantly on the perceived risk. This paper investigates the difference between objective and perceived risk and highlights some critical issues. The aim of the study is to calibrate opportune policies, which allow addressing the most appropriate seismic risk mitigation options with reference to current levels of perceived risk. We propose the introduction of a Seismic Policy Prevention index (SPPi). This methodology is applied to a case-study focused on a densely populated district of the city of Catania (Italy).

**Keywords:** seismic vulnerability; urban areas; objective risk; perceived risk

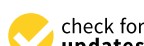



## 1. Introduction

Risk analysis represents a complex and multidisciplinary field. It can be split into a "hard" approach, typically attributed to quantitative sciences based on mathematical and probabilistic models, and a "soft" approach based on qualitative investigations. Each approach captures a different and partial aspect of the complexity and multidimensionality of risk [1]. Proper risk assessment based on scientific foundations is a necessary but not always sufficient condition for sound policy decisions. In risk analysis there is an increasing need to correlate these two different approaches. The proper balance between the scientific and "objective" component of risk and the "subjective", value-based, ethical component is probably one of the main challenges facing contemporary democratic governments and their regulatory systems. Some important insights in social sciences about risk perception find direct application in the domain of risk management and planning strategies, thus highlighting how it assumes an important role in risk regulation and management [2].

While the objective risk can be related to the actual state of vulnerability of a building, the perception of risk by different actors has an inherently subjective nature.

The vulnerability assessment of existing buildings at an urban scale has been developed by many authors by means of simplified procedures based on empirical or analytical assessment approaches. Several authors have proposed models for the seismic vulnerability assessment of urban areas around the world. Pavic et al. focused on exposure models in order to quantify the building stock in terms of structural characteristics, spatial location and occupancy [3,4]; Salazar and Ferreira proposed a simplified methodology, composed of both quantitative and qualitative parameters and integrated on a GIS platform, able to map vulnerability and damage scenarios after earthquakes of various intensities [5]; Ferreira et al. proposed a simplified methodology based on the evaluation of eight parameters associated with different factors that affect the seismic response of the building, namely its structural features, foundation conditions, and position within the urban mesh [6]; Greco et al. focused on the recognition of a structural modulus of unreinforced masonry buildings studied according to different geometrical layouts representative of isolated or aggregate buildings: the nonlinear static analyses, performed by applying an innovative macro-element approach, allowed the assessment of the seismic vulnerability of typically unreinforced masonry buildings [7]; Zhai et al. introduced a GIS-based seismic hazard prediction system for urban earthquake disaster prevention planning and developed incorporating structural vulnerability analysis, program development, and GIS approach [8]; Vargas-Alzate et al. incorporated the non-linear dynamic analysis of a structure's response and uncertainties related to the loads, the geometry of the buildings, the mechanical properties of the materials and seismic action [9]; Greco et al. proposed a methodology able to combine information and tools coming from different scientific fields in order to reproduce the effects of a seismic input in urban areas with known geological features and to estimate the entity of the damages caused on existing buildings [10]; and Fischer et al. further developed the previous methodology for addressing damage scenarios and risk assessment in large urban systems, such as metropolitan areas [11].

On the other hand, risk perception can be seen as a subjective assessment of the probability that a specific hazard may occur, and as the measure of how concerned individuals are about the consequences [12]. A similar interpretation is given in [13], in which the perceived risk is defined as "subjective opinions of people about the risk, its characteristics, and its severity including multiple factors: individual's knowledge of the objective risk, individual's expectations about his or her own experience to the risk, and his or her ability to mitigate or cope with the adverse events if they occur".

Dowling and Staelin [14] define perceived risk as a partial and subjective assessment of objective risk and point out in many cases a clear difference between perceived subjective risk and objective risk assessment. However, they state that it is generally the perceived risk, more than the objective one that motivates individuals to engage in particular behavioral patterns, ameliorative to the threat of danger. According to Slovic [15], risk perception depends on several factors: the controllability of the hazard, the catastrophic potential, and the degree of uncertainty. This perception plays an important role in how individuals choose to mitigate it. For example, if individuals estimate a low risk, they will be less likely to act to reduce their exposure [16]. A study conducted in L'Aquila after the 2009 earthquake found that low risk perception inhibited the propensity to develop contingency plans, and even unwarranted reliance on buildings constructed with sack masonry underestimated the need for structural intervention with seismic retrofitting and improvement of buildings [17].

Risk perception is a critical element in the socio-political context in which policymakers operate, as it determinates politico-economic and social actions aimed at reducing the possible consequences resulting from hazardous events. Some researchers have shown that mitigation measures, both structural and non-structural, can be rejected by society if applied before investigating public perception of risk [18], because risk perception influences safety behaviors [19]. This behavior is understandable, since people often contrast what they do not know or understand and tend to believe that such measures are not necessary. These

beliefs can be changed when the contingency between mitigation actions and positive outcomes of an action is demonstrated [20].

Risk perception is also dependent on the type of hazard, as each one has peculiar features and affects people differently. The proximity of the hazard, its intensity, and the rate of recurrence are some of factors influencing how people interpret risk. The rate of occurrence of the earthquake event varies the risk perception significantly, as shown by Saito et al. [21]: this perception increases significantly when the probability of occurrence is presented in an interval of 20–30 years for a population aged 20–40 years, or 10 years for a population aged 50–60 years.

Perceived risk also depends on the institutional ability to communicate concepts that are difficult to understand because they are loaded with technical-scientific content, such as the low probability of occurrence of a high-magnitude seismic event. To this end, Savadori et al. showed that a comparison with different and more familiar hazards significantly improved risk perception and sensitivity to information on prevention issues [22].

The perception of earthquake risk is strongly related to these factors, but it also depends on demographic (gender, age, education), socio-economic, socio-cultural and psychological factors. Partly, it depends on previous experiences of seismic events [23], which may alter individual perceptions of risk and, in turn, precautionary decisions and behaviors about the future. Understanding how risk is perceived by the public is an important step in assessing the vulnerability of a community [24,25].

Studies in the area of risk perception are still in their first steps, especially concerning its actual role within urban policy and decision-making processes. According to Pidgeon [2], the adoption of the perceptual dimension of risk in urban policies has two relevant implications of a normative and epistemological nature. In the first case, incorporating perceptions of risk into urban policies increases public participation on decisions regarding individual and collective safety, implying a greater willingness to pay for one's safety than dangers evoking feelings of fear [26]. The latter attitude, which could be called "propensity to change", also has positive effects on overall risk management, especially in communities with a high level of cohesion. Indeed, when people observe preventive "adjustment" ' behavior by neighbors or other community members, they tend to emulate them [27].

The literature suggests that the predisposition to adopt risk-reducing behaviors is a positive measure of the overall amount of perceived risk. Martin et al. [28] explore the mediating relationship [29] between "risk perception" variables (understood as knowledge of danger, direct experience with hazardous events, and individual responsibility), hazard mitigation and reduction behaviors, and assert that risk perception has a significant mediating effect, conditioned particularly by individuals' knowledge of what they believe, what they know about danger and their sense of responsibility in their willingness to protect themselves, their property, and their families. O'Connor et al. [30] propose a model in which risk perception and hazard knowledge are related to behavioral intentions regarding actions to be activated to cope with climate change.

Crescimbene et al. [31] conducted a study on the perception of risk factors: vulnerability, exposure, hazard, institutions and community, and earthquake phenomenon through a web-administered questionnaire with random sampling. A comparison is made between the hazard as it is mapped "by the law", i.e., the hazard expressed by the four seismic zones identified in Italy, MPS04/OPCM 3274 and [32], and the perceived hazard. The definition of an appropriate reference value of perceived hazard for the four Italian seismic zones made it possible to highlight the difference between them.

The objective of the traditional hazard analysis is to guide effective mitigation and adaptation interventions in urban systems. It involves adopting urban strategies to reduce the hazard factors identified through an analysis of its physical-quantitative components and developing appropriate measures to minimize the damage that seismic impacts can cause [33] due to system failure and inability to respond.

If the expected damage is a function of the number of potential victims, the most commonly adopted strategies for reducing the effects of a potential hazard event include

reducing exposure to seismic hazards by reducing the residential/functional density of a city and reducing the physical/functional seismic vulnerability of the components that insist on it.

This approach imposes an assumption: seismic events cannot be controlled but the consequences, with respect to the events, can be effectively influenced [20]. The adoption of the perceptual dimension in risk analysis and assessment depends on the indicators that can define and capture the determinants of risk perception, as it may differ from hazard to hazard and from community to community. Generally, the perception of risks and mobilization against them takes place only when the everyday life spheres of individuals or social groups, and thus their local context, are closely involved. The social aspect of risk is often context-specific and therefore local in nature. The most relevant risks, in social terms, are such that, even when conditioned by causes of a global order, their manifestation reveals connotations that have specificities peculiar to the various contexts [34]. For this reason, the approach to this type of analysis is community-based.

The inclusion of perceptual factors in the traditional representation of risk means considering the assessment of risk by lay people. The mismatch between "expert" and "lay" risk assessment makes it more difficult to reach consistent and shared decisions on strategies to reduce and mitigate risk in environmental planning processes.

This paper introduces a method that, through the investigation of the gap between objective and perceived risk, aims to understand the critical elements that prevent the adoption of relevant seismic risk-mitigation measures. The method makes it possible to investigate these critical issues and intervene with appropriate policies to fill this gap. The end point of our research is to calibrate policies by identifying an index, called the Seismic Prevention Policy index (SPPi), that allows us to address the most appropriate seismic risk mitigation options, after ranking them, in relation to the current levels of perceived risk.

The methodology is applied to a densely populated neighborhood of the city of Catania (Italy), in which a very high-risk situation emerges due to the combination of several factors. First of all, Catania, and all the oriental coast of Sicily, is an area with very high seismic risk due to its tectonic and geological conditions and the frequency of occurrence of earthquakes that have historically affected its territory.

In addition, since the majority of the buildings in the considered district have been there for more than 60 years, the level of seismic vulnerability of the built heritage is high due to its deterioration over time. Furthermore, the risk is strongly influenced by the ageing characteristics of the population (which represent a constantly growing trend throughout the Mediterranean area) and the high population density.

The rest of the paper is organized as follows. In Section 2 the conceptual framework of the followed risk approach is sketched, together with the adopted methodological tools; in Section 3 the definitions of both objective and perceived risk are introduced, the chosen case study is addressed in detail and the main risk indices are built from the data. Finally, in Section 4, the main results of our analysis are presented and possible policy strategies are discussed. Section 5 closes the paper, drawing some conclusions.

## 2. Conceptual Framework and Methodology

The present research aims to balance the "hard" and "soft" approach of risk analysis (Figure 1). The framework presents two main concepts: "objective risk" and "perceived risk". The primary objective is to compare the two approaches to risk analysis in order to assess how much the community's perception of risk can influence the application of urban risk-reduction policies, depending on objective levels of measured seismic risk. Objective risk assessment is analyzed through three components, represented by vulnerability, exposure and hazard [35]. Perceived risk assessment is identified by five determinants: subjective knowledge, hazard experience, self-efficacy, social and personal factors and nature and features of disasters [28,36].

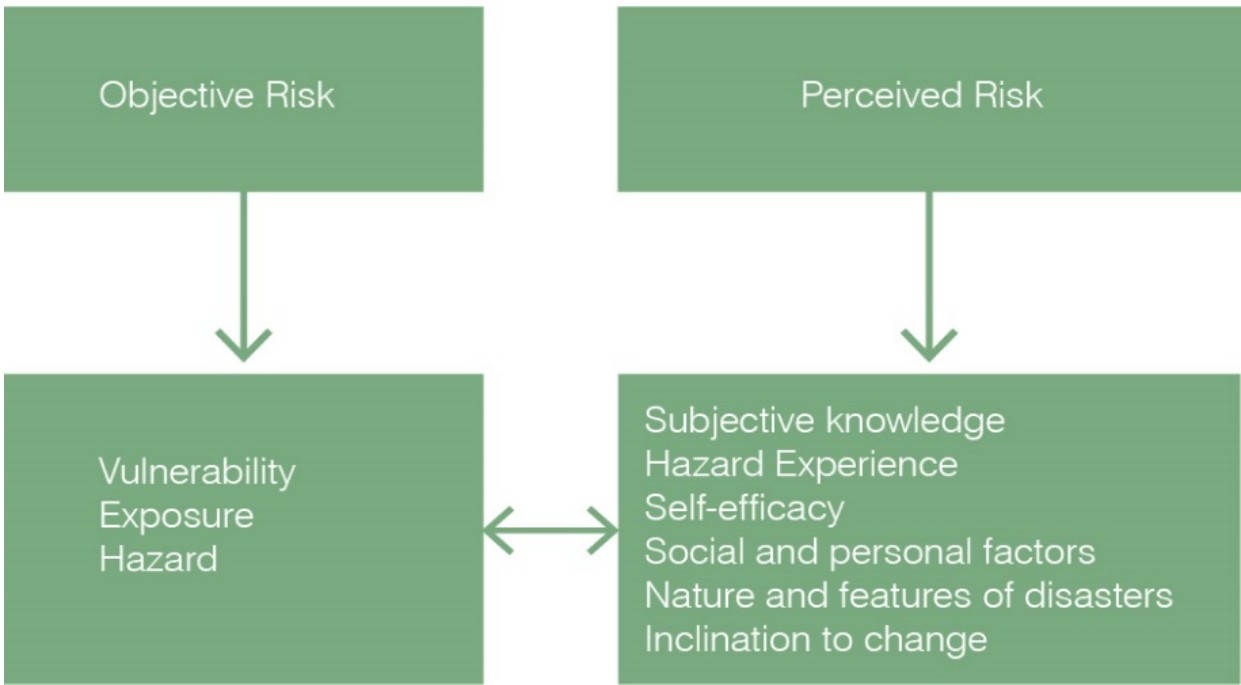

**Figure 1.** Conceptual Framework showing objective risk assessment model and determinants of perceived risk.

The analytical dimension on the neighborhood scale that defines the most appropriate community level for investigating hazard perception, which may vary from community to community, depending on the type of hazard. Based on this approach, the research aims to modulate urban transformation-prevention policies according to the distance-difference between existing perceived risk and objective risk.

The proposed method can be exploited in the planning stage of the strategic prevention objectives of the urban plan. The purpose is to identify areas of urban transformation-prevention, and to provide a tool that can effectively guide urban policies especially in those areas where the objective risk is greater than the perceived risk. In this regard, it is also necessary to assess neighborhood residents' propensity to change in order to provide appropriate incentives for earthquake risk mitigation.

The tool introduces, for the first time, the subjective-perceptual dimension in prevention planning and urban transformation, emphasizing the indispensable contribution of the community to the implementation of urban seismic prevention policies. Figure 2 shows the general methodology adopted in the proposed research. It is divided into two phases: analysis and mapping of objective risk and analysis and mapping of perceived risk. The traditional "hard" approach to risk assessment adopts a well-known model of the literature, i.e., the Crichton model [35]; and a "soft" approach unique to social research, based on the adoption of a survey that investigates the components of risk perception according to the models already mentioned in the previous section.

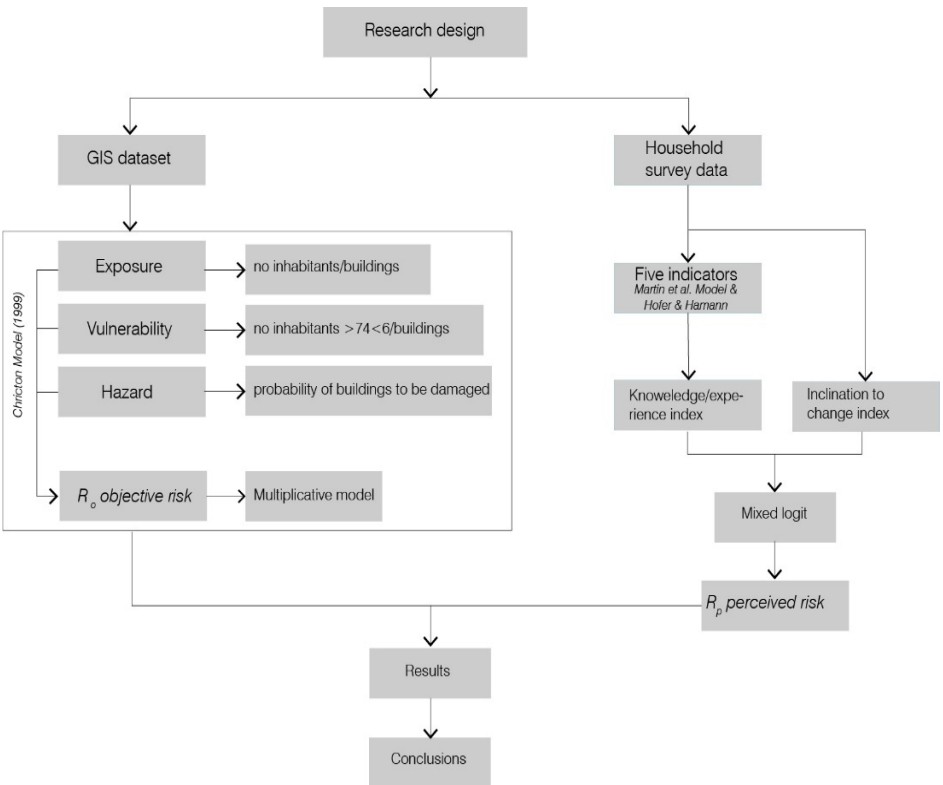

**Figure 2.** General methodology.

## 3. Material and Method

### 3.1. Objective and Perceived Risk

As anticipated, the assessment of objective risk in the urban context is calculated with reference to Crichton's model [35]. According to this model, risk ($R_o$) depends on three elements: hazard (H), vulnerability (V), and exposure (E). These factors may assume different meanings depending on the nature of the event producing harm (e.g., earthquake, epidemics, flooding, climate change) and on the size of the considered territory. In the present study, with the aim of evaluating the seismic risk referred to a portion of an urban area, such as a district, the factors of vulnerability, exposure and hazard can be appropriately defined.

More precisely, since the intensity of an earthquake does not significantly vary within the district (although it could be related to the nature of the foundation soil), hazard is defined as the probability that a building may suffer damage as a result of the seismic event and has therefore been related to its (seismic) vulnerability. Exposure is parameterized according to the number of inhabitants in each building unit. Inhabitants' vulnerability is expressed in relation to the percentage of population over 74 years old and under 6 years old [37] for each building unit.

Seismic vulnerability is the propensity of a building to be damaged by seismic events of a given intensity. This vulnerability strongly depends on the structural typology, on the mechanical properties of the constitutive material and also on the maintenance level of the building. In fact, it is well known that the resistance of construction materials decreases over time due to material quality, corrosion phenomena caused by atmospheric agents, stress state, rate and possible repetitiveness of load application. Other causes of vulnerability in structures can be linked to the presence of concentrated structural failures or damage caused by previous seismic events. A detailed evaluation of the seismic vulnerability of a building would therefore require an accurate level of knowledge of the structure and of its maintenance conditions, and this can only be pursued following extensive structural investigations with 'in situ' tests. This approach cannot be obviously applied on an urban

scale, where it is essential to refer to quick but at the same time reliable methods to assess seismic vulnerability of buildings. A commonly adopted approach is based on statistical methods according to which a vulnerability score is assigned to buildings. Numerous studies have been developed for this purpose in different countries [5,38–46].

The method for the detection of the vulnerability of existing buildings that has been applied in this study refers to the procedure adopted by the GNDT (National Group for Earthquake Defense) first proposed by Benedetti and Petrini [47], which is based on the compilation of appropriate forms specifically prepared for masonry or reinforced concrete structures, through the identification of some fundamental reference parameters.

The vulnerability of reinforced concrete buildings is calculated as a function of seven parameters (age, resistant system, average normal tension of 1st level columns, plan regularity, 1st level infill type, non-structural elements, building position and foundations), to which the corresponding scores are attributed according to the values assumed by the indicators themselves, obtaining a vulnerability index between 25 and 100 (OPCM-DRPC No.3105/2001). The vulnerability of load-bearing masonry buildings is calculated by the means of nine parameters (efficiency of connections between walls, quality of masonry, location of the building and foundation soil, conventional strength, horizontal structures, planimetric configuration, roofing, nonstructural elements, and state of affairs) of which the scores given to each class are added together, obtaining a seismic vulnerability index between 0 and 100 (OPCM-DRPC No.3105/2001). Vulnerability indices for reinforced concrete and load-bearing masonry buildings are normalized against their maximum value.

The objective risk for the $j$-th building ($j = 1, 2, \ldots, n$) can be defined adopting the Crichton's multiplicative model [35], which takes into account hazard, exposure and vulnerability, and is expressed by the formula:

$$(R_o)_j = \frac{(V_b)_j}{\max[(V_b)_j]} \frac{(V_{6-74})_j}{\max[(V_{6-74})_j]} \frac{(E_{ab})_j}{\max[(E_{ab})_j]} \tag{1}$$

where:

- $(V_b)_j$ is the probability that a building may suffer damage as a result of an earthquake, parameterized in relation to the seismic vulnerability of the individual building;
- $(V_{6-74})_j$ is the percentage of inhabitants under 6 years old and over 74 years old for each building;
- $(E_{ab})_j$ is the total number of inhabitants for each building.

All quantities considered in Equation (1) are normalized to their maximum values for each building. In addition, the value of the obtained objective risk index is, in turn, normalized to the range 0–1 (min-max normalization).

Based on two reference models by Martin et al. [28] and Hofer and Hamann [36], some useful indicators for defining risk perception and propensity to change can be selected. Such indicators are subjective knowledge, previous experience with danger, self-efficacy [28] and personal and social factors, nature and characteristics of disasters [36], respectively. Concerning the perceived risk, the relevant literature has shown that those with greater experience of a given hazard have greater awareness and knowledge of that hazard and adopt alternative strategies to deal with it through the adoption of risk-reducing behaviors [48,49]. Subjective knowledge is also based on direct experience (involving evacuation from homes, loss of property, temporary stay out of one's home, or other similar experiences) and indirect experience (word of mouth, information from relatives and friends, etc.) with respect to a particular type of hazard. Personal and direct experience has a major impact on the recognition of risk and the propensity to protect oneself and one's assets. Past experiences are the starting point for the formation of subjective knowledge of risk and can be useful in influencing the decision-making process [15]. Sattler et al. found that individuals tend to base their risk awareness of future hazards on the extent of potential

harm and psychological stress caused by past experiences [50]. The authors found that people perceive past experiences as indicators of potential future negative experiences.

Perceived risk can thus be analyzed by administering a questionnaire to the resident population. The questionnaire is divided in relation to the five indicators of the Martin et al. and Hofer and Hamann model [28,36], which are necessary to define the perception of earthquake risk in terms of individual knowledge and experience. To each of the five indicators, representing the characteristics of individual knowledge and experience of risk: subjective knowledge, hazard experience, self-efficacy, social and personal factors, nature and features of disasters, correspond a certain number of questions, as suggested by the literature on the subject. In addition to the five indicators, the survey aims to define the inclination to change index, which identifies people's willingness to adopt "ameliorative" behaviors also in light of the presence of incentives that facilitate access to resources needed for earthquake adaptation. Each question has four response options that are associated with a score ranging from 1 (very low) to 4 (high). The summation of the scores for each respondent defines the summary value of the indices of knowledge and experience of risk and inclination to change, respectively, according to the following equations:

$$I_{KE} = \sum_{i=1}^{m} i_{KE}/m \tag{2}$$

$$I_{IC} = \sum_{i=1}^{k} i_{IC}/k \tag{3}$$

in which:

- $i_{KE}$ represents the $m$-th variable of the risk knowledge and experience index;
- $i_{IC}$ represents the $k$-th variable of the inclination to change index;

while $m$ and $k$ represent the number of variables useful to define the risk perception and inclination to change index, respectively.

The assessment of perceived risk, $R_p$ (Risk perceived) for the $j$-th building ($j$ = 1, 2, . . . , $n$) can thus be evaluated through the formalism of the multinomial logit models (MNL) [51]:

$$(R_p)_j = e^{w_{KE} <I_{KE}>_j + w_{IC} <I_{IC}>_j} / \sum_{i=1}^{n} e^{w_{KE} <I_{KE}>_i + w_{IC} <I_{IC}>_i} \tag{4}$$

where:

- $< \ldots >_j$ represents an average over all the respondents belonging to the $j$-th building
- $w_{KE}$ represents the weight of the knowledge and experience of risk index;
- $w_{IC}$ represents the weight of the inclination to change index.

The actual assessment of perceived risk thus depends also on these weights, whose values could be estimated through appropriate surveys. In the absence of such surveys, without loss of generality, the same value (in this case 1/2) can be adopted for both weights.

Finally, the ratio between the estimated value of objective risk (1) and perceived risk (4) defines a policy indicator for the $j$-th building, referred to below as the Seismic Prevention Policy Index (SPPi):

$$(SPPi)_j = (R_o)_j/(R_p)_j \tag{5}$$

which can be further averaged over all the $n$ buildings obtaining:

$$(SPPi)_{TA} = \sum_{j=1}^{n} (SPPi)_j/n \tag{6}$$

which will be the $(SPPi)_{TA}$ Seismic Policy Prevention Index of the urban transformation area.

### 3.2. Case Study

A district in the Catania Metropolitan Area, located in Sicily, Italy (Lat. 37.30 North, Long. 15.07 East), is considered as the case study (Figures 3 and 4) to show the potentiality

of the methodology described in the previous section. The district is characterized by high per-capita volumes, high residential density, absence of open spaces, buildings built prior to earthquake regulations, and an ageing index among the highest in the municipal area. The area investigated measures 10.71 ha and houses a population of 3.604 individuals, with a density that stands at values of 340 ab/ha; as shown in Figure 3, this means that this neighborhood has the characteristic of containing in a very small area the equivalent of the population of residents in other wider districts.

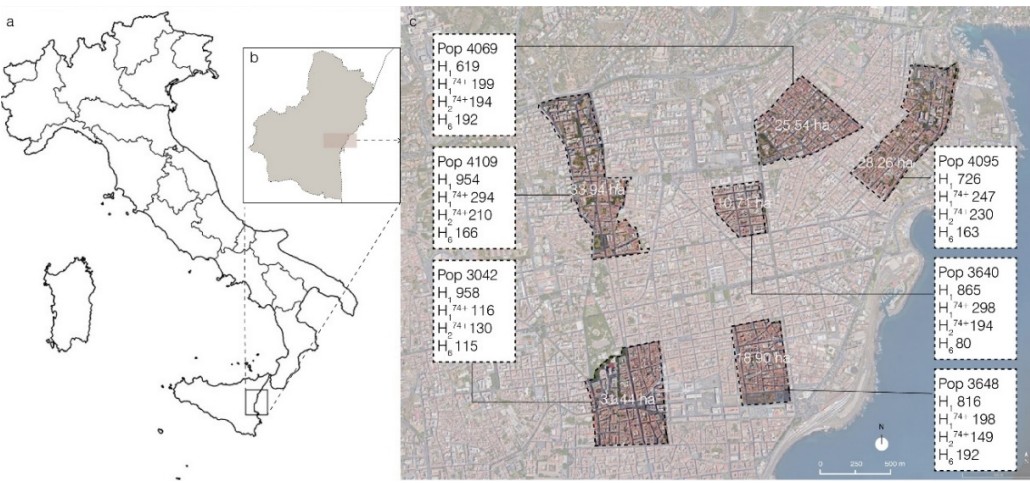

**Figure 3.** (**a**) Geographical identification of the Catania Metropolitan Area (**b**) Perimeter of the Catania Metropolitan Area and (**c**) Location of Catania's districts and possible application of the methodology adopted.

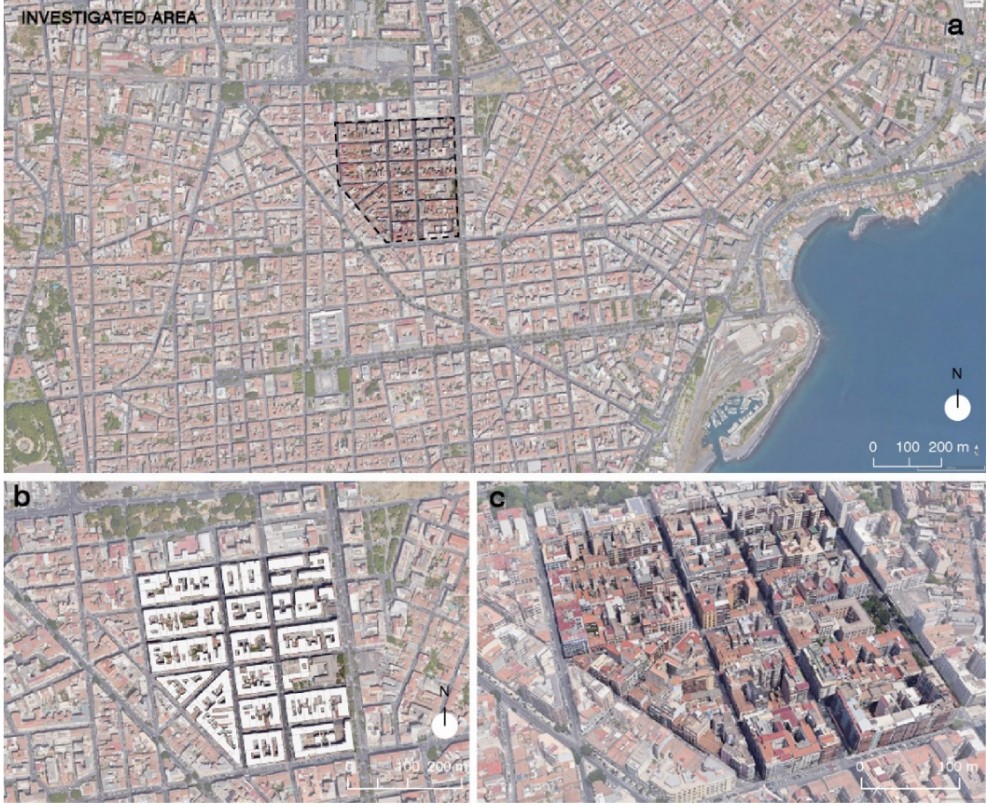

**Figure 4.** Study area: (**a**) Satellite image of the study area; (**b**) Identification of the 164 buildings surveyed; (**c**) Three-dimensional view of the survey area.

The considered district is defined as a residential area with a functional mix in the city's General Regulatory Plan. Figure 4 identifies the investigated area in the two- and three-dimensional satellite images (Figure 4a,c); it is also visible the building fabric consisting of 164 buildings (Figure 4b).

A total of 72% of the buildings are constructed of reinforced concrete, compared with the remaining 28 percent in load-bearing masonry.

The area is characterized by an orthogonal mesh texture, with streets ranging in size from 6.50 to 12.00 m. On average, buildings have a height of 20.1 m, resulting in an average total number of 6–7 stories. Contrary to current regulations, the height of buildings in the investigated area frequently exceeds the width of streets. Moreover, the buildings falling in the investigated area, having been built before Catania was classified as a seismic zone, do not comply with the current seismic regulations.

The population survey shows that the younger population cohort (0–14 years old) is smaller than the older population one, highlighting a regressive population structure. In fact, compared with a population over 64 years old ($Pop^{64+}$) of 1066 individuals, the younger population ($Pop^{15-}$) consists of 353 elements (Table 1).

**Table 1.** Population and built-up area data, land use and road parameters of the investigated area.

| Description | Symbol | Value | | Unit |
|---|---|---|---|---|
| Area | $A_{site}$ | 10.71 | | ha |
| **Population** | | | | |
| Population | Pop | 3604 | | - |
| Population density | $Pop/A_{site}$ | 336.50 | | Inh/ha |
| Households | H | 1815 | | - |
| Medium age | Ma | 49.59 | | - |
| Median age | Mea | 51 | | - |
| Population $^{64+}$ | $Pop^{64+}$ | 1066 | | - |
| Population $^{15-}$ | $Pop^{15-}$ | 353 | | - |
| Population $^{6-}$ | $Pop^{6-}$ | 111 | | - |
| Population $^{74+}$ | $Pop7^{4+}$ | 598 | | - |
| | | | Rate | % |
| Ageing Index | Ai | - | $Pop64+/Pop^{14-}$ | 302 |
| One member households | H1 | 865 | $H_1/Pop$ | 24 |
| One member 74+ households | $H_1^{74+}$ | 298 | $H_1^{74+}/Pop$ | 8 |
| Two members households | $H_2$ | 852 | $H_2/Pop$ | 24 |
| Two members households 74+ | $H_2^{74+}$ | 194 | $H_2^{74+}/Pop$ | 5 |
| Two members households 18− | $H_2^{18-}$ | 48 | $H_2^{18-}/Pop$ | 1 |
| Households 6+ | $H^{6+}$ | 80 | $H^{6+}/Pop$ | 2 |
| Foreign residents | Fr | 199 | $Fr/Pop$ | 6 |
| **Built-up area data** | **Symbol** | | **Value** | **Unit** |
| Buildings Number | B | | 164 | - |
| Total buildings volume | $V_{bds}$ | | 1,218,243 | m$^3$ |
| Buildings density | $V_{bds}/A_{site}$ | | 113,748 | m$^3$/ha |
| Total residential buildings volume | $Vr_{bds}$ | | 993,886 | m$^3$ |
| Average building height | $h_{wtd}$ | | 20.1 | m |
| Floor average number | $F_{av}$ | | 6 | - |

**Table 1.** *Cont.*

| Description | Symbol | Value | | Unit |
|---|---|---|---|---|
| Building's total gross floor area | $A_{bldg}$ | | 56,089 | $m^2$ |
| Dwelling units | Du | | 2.464 | - |
| Soil use fraction | Symbol | | Value | Unit |
| Floor area ratio | FAR | | 52.37 | % |
| Impervious surface area | $A_{imp}/A_{site}$ | | 99 | % |
| Pervious surface | $A_{per}/A_{site}$ | | 0.09 | % |
| Street parameters | | | | |
| Street width | W | | 6.50–12.00 | m |
| Street aspect ratio | H/W | | 0.85–3.00 | m |
| Sidewalks width | W | | 0.80–1.20 | m |

The ageing index, given by the ratio of the population over 64 to the population under 15, is 302, double the city average, which stands at 158.1 (Istat, 1 January 2019). The ageing of the population becomes an even more alarming vulnerability factor when placed in the analyzed context, in which the number of members per family is decreasing. In particular, the single-component family with a member over 74 years old accounts for more than 40 percent of the settled inhabitants in at least 5 percent of the building units in the investigated area.

The 164 housing units in the investigated area house a total of 2.464 apartments. An average of 1.46 people live per apartment, highlighting a process of under-utilization of the building stock, made even more evident by the value of residential volumetry per capita averaging 380.17 $m^3$/inhabitant, compared to the 80–100 $m^3$/inhabitant indicated in DM 1444/68 as the optimal value.

The calculation of residential volumetry was developed by eliminating from the calculation one floor, corresponding to the ground floor and intended for commercial activities, or pertaining to buildings as technical rooms or with storage or shed use. Occurrences in the range 225–450 $m^3$/ab represent 45% of the cases. Values from 5 to 33 times higher (450–2636 $m^3$/ab) than those considered optimal in DM 1444/1968 are identified in 20% of the occurrences. The under-utilization of the building stock aggravates the neighborhood's vulnerable conditions due to the lack of maintenance work, including routine maintenance, required to upgrade the buildings.

In order to evaluate the objective risk for all the buildings of the neighborhood, the latter need to be classified according to their typological-constructive characteristics (OPCM-DRPC No.3105/2001). The considered district is divided into four sectors (A-B-C-D), within which blocks (with a numerical code) and individual building units (with alphabetical code) are, in turn, identified (Figure 5).

Merging this information, seventeen building types can be further identified through an alphanumeric code, then to each of them a seismic vulnerability value $V_b$ can be assigned. Just to present an example of vulnerability calculation, let us consider a typical, open-court, insulated reinforced concrete building (code RC-oc I8), for which the corresponding parameters are shown in Table 2 and can be used to evaluate $V_b$ through the following formula:

$$V_b = V_m + a \sum_i k_i = 12 + 88 \times 0.425 = 49.4 \tag{7}$$

where $V_m$ is the average vulnerability value for this type of building and *a* is another tabulated parameter.

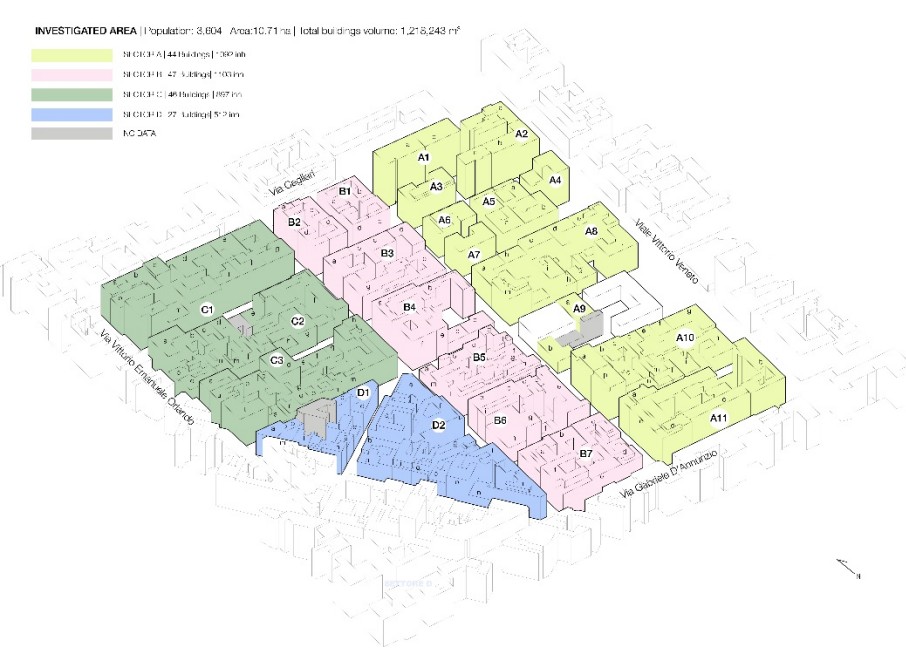

**Figure 5.** Study area with the identification of four sectors, blocks and building units.

**Table 2.** Calculation of the vulnerability level for a typical concrete, closed-court, insulated, 8-story building (RC oc-I8).

| Parameters $k$ | Typology: RC oc-I8 | |
| :---: | :---: | :---: |
| | Description | Value $\epsilon$ [0, 1] |
| 1.1 | Construction date | 0.1875 |
| 1.2 | Resisting system type | 0.125 |
| 1.3 | Mean column normal stress at 1° level ($\sigma$ = Np $*$ q $*$ A/Sp) | 0.1875 |
| 1.4 | Regularity | $-0.2$ |
| 1.5 | Infill typology 1° level | 0.125 |
| 1.6 | Non-structural elements | 0 |
| 1.7 | Location and soil condition | 0 |
| $\sum k_i=$ | 1.1 + 1.2 + 1.3 + 1.4 + 1.5 + 1.6 + 1.7 | 0.425 |

Following an analogous procedure, the vulnerability is estimated for all the seventeen classes and the obtained values, normalized to the maximum, are reported in Table 3. Then, once calculated $(V_b)_j$ for each building $j$-th of the various types, the further information about $(V_{6-74})_j$ and $(E_{ab})_j$ can be extracted from our survey dataset and allow us to apply Equation (1) for the final evaluation of the objective risk $(R_o)_j$.

The assessment of perceived risk $(R_p)_j$ is developed according to Martin and Hofer and Hamann's models [28,36] by means of the administering of a questionnaire. The questionnaire administration phase was preceded by a series of educational meetings with students at a Secondary School located in the investigated area. The meetings, had as their subject the knowledge of the seismic phenomenon and the evaluation and perception of seismic risk in the city of Catania allowed to develop the skills necessary for the administration of the questionnaire. Students then conveyed a more articulated questionnaire to their families through the Google Form platform. The sample selection expresses the connection to the neighborhood by means of indicating the respondent's area of residence. Out of a total of 250 questionnaires administered, only questionnaires whose respondents reside in the investigated area were selected. The sample was chosen randomly and consists of 118 respondents who reside in the investigated neighborhood. The indicators defining the

Risk Knowledge and Experience Index were then analyzed through descriptive statistics to obtain information on the performance of each of them and explain their importance in defining risk reduction measures for earthquake prevention policies. The details of the analysis are shown in Table 4 and synthetized as follows:

**Table 3.** Vulnerability calculated for each of the 17 typological-structural categories in the survey area.

| Typology | Number of Buildings | Vulnerability $V_b$ | Typology | Number of Buildings | Vulnerability $V_b$ |
|---|---|---|---|---|---|
| RC_lA11 | 4 | 0.14 | RC_oc1A9 | 7 | 0.74 |
| RC_l I10 | 1 | 0.32 | RC_oc2A8 | 8 | 0.74 |
| RC_li A9 | 13 | 0.32 | RC_oc2A9 | 4 | 0.74 |
| RC_li A11 | 9 | 0.32 | RC_l A9 | 22 | 1 |
| RC_cc A8 | 2 | 0.32 | MA_mh3 | 11 | 0.26 |
| RC_oc A7 | 22 | 0.35 | MA_mh5 | 10 | 0.34 |
| RC_l A6 | 15 | 0.35 | MA_ml3 | 16 | 0.67 |
| RC_oc I8 | 4 | 0.58 | MA_ml5 | 7 | 0.79 |
| RC_oc1 A8 | 9 | 0.74 | | | |

**Table 4.** Descriptive statistics of risk perception indicators.

| Weights | Classes | Frequency | % |
|---|---|---|---|
| Subjective knowledge | | | |
| 1 | Very high | 1 | 0.8 |
| 0.8 | High | 16 | 13.5 |
| 0.6 | Moderate | 63 | 53.5 |
| 0.4 | Low | 21 | 17.8 |
| 0.2 | Very low | 17 | 14.4 |
| Self-efficacy | | | |
| 1 | Very high | 6 | 5 |
| 0.8 | High | 26 | 22 |
| 0.6 | Moderate | 51 | 43 |
| 0.4 | Low | 16 | 14 |
| 0.2 | Very low | 19 | 16 |
| Hazard experience | | | |
| 1 | High | 22 | 18.7 |
| 0.5 | Moderate | 88 | 74.6 |
| 0 | Low | 8 | 6.7 |
| Nature and features of disasters | | | |
| 1 | Very high | 42 | 35.6 |
| 0.8 | High | 15 | 12.7 |
| 0.6 | Moderate | 30 | 25.4 |
| 0.4 | Low | 20 | 17 |
| 0.2 | Very low | 11 | 9.3 |

*Subjective knowledge*: Descriptive analysis found that only 14% of respondents report having high or medium-high knowledge of the seismic safety characteristics of their property (namely building code, construction type) and the seismicity of their city of residence.

*Self-efficacy*: 30% of respondents do not take preventive measures in terms of seismic certification for property rental and have not carried out structural interventions as a result of seismic events.

*Hazard experience*: Nearly all respondents have had direct or indirect experience with earthquakes, and only 6% have had no experience with earthquakes.

*Nature and features of disasters*: On average, 48% of respondents say they understand the seismicity characteristics of their area of residence and the possibility that a high-magnitude earthquake event could occur again in the city of Catania. They also understand the amplification characteristics of earthquake effects in man-made settings (Table 4).

A multiple regression model was applied to ascertain the factors influencing the experience and risk knowledge index. The results of the multiple regression, presented in Table 5, suggest that the model fits well the input data (F = 13.36, *p*-value = 0.000), allowing us to say that the indicators used are suitable to explain the model.

**Table 5.** Multiple linear regression results.

| Model | Coefficient B | Std. Error | *t*-Value | *p*-Value |
|---|---|---|---|---|
| Constant | 1.613 | 0.150 | 10.728 | 0.00 |
| Age | 0.112 | 0.028 | 3.888 | 0.00 |
| Education | 0.128 | 0.046 | 2.762 | 0.01 |
| Occupation | 0.064 | 0.015 | 4.172 | 0.00 |
| House ownership | 0.130 | 0.061 | 2.138 | 0.03 |
| Model summary | $R^2 = 0.3212$ | | | |
| ANOVA | F = 13.36867 | *p*-value = 0.00 | | |

Regarding the relationship between age and the $I_{KE}$ index, there is a positive relationship between the two variables. The respondents were categorized into five groups: 18–25, 26–35, 36–50, 51–65, and over 65. The results show that age has a positive influence on the index, certainly conditioned by previous experience with earthquakes that have struck the city. This implies that older people will have higher index values when compared to younger respondents. An individual's level of education may influence the risk knowledge and experience factors, as evidenced by the positive relationship between the two variables in the model. Employment status also influences $I_{KE}$, but to a lesser degree than the other variables.

To each respondent is given a score for the Index of Knowledge and Experience of Risk ($I_{KE}$) and the Index of Inclination to Change ($I_{IC}$). The $I_{KE}$ is calculated on a scale of 1 (very low) to 4 (high) and the $I_{IC}$ on a scale of 1 (low) to 3 (high). The scores are calculated according to formulas 4 and 5.

The range of variation in the risk knowledge and experience index among respondents is between 2 and 3.61. Seventy-five percent of the respondents have a risk knowledge and experience index below 3.30. The range of change in the propensity to change index is between 1.63 and 3.00. Seventy-five percent of the respondents have a propensity to change index that is less than 2.50. Only the last quartile has values above 2.50.

As made evident in Figure 6, there is a positive correlation between the indices, indicated by the value of $R^2 = 0.142$, which is an expression of the greater propensity to activate attitudes and improved practices for individual and collective safety the higher the Risk Knowledge and Experience Index (Table 6).

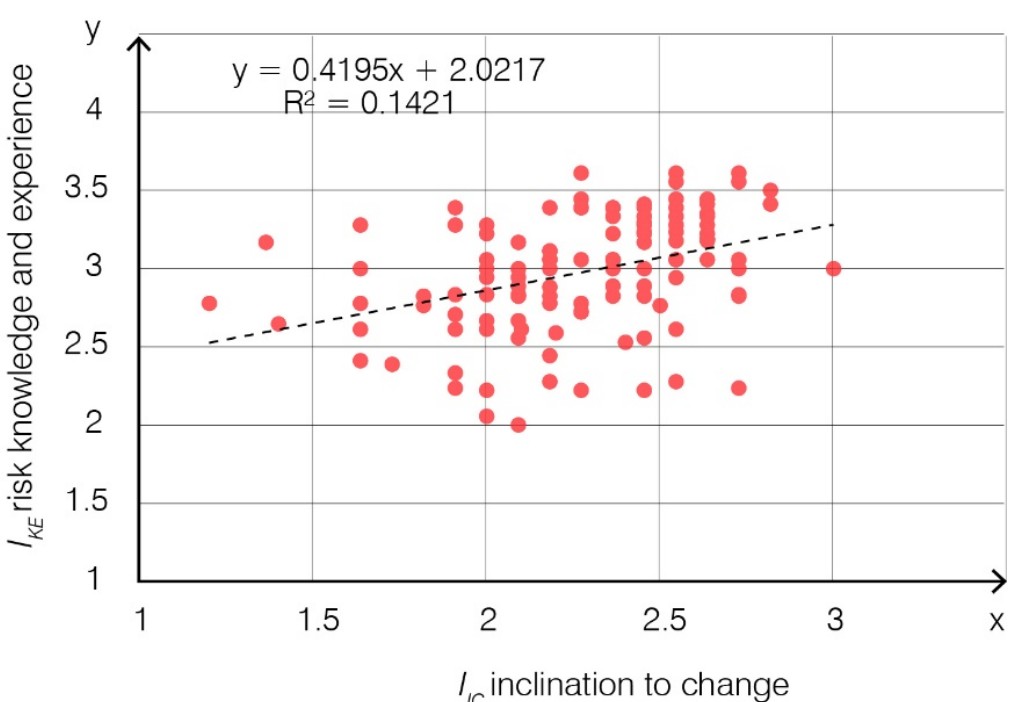

**Figure 6.** Correlation diagram between the Indices of Knowledge and Experience of Risk and Inclination to Change.

**Table 6.** Pearson correlation between the Knowledge and Experience of Risk ($I_{KE}$) and Inclination to Change ($I_{IC}$) indices.

|  |  | $I_{KE}$ |
|---|---|---|
| $I_{IC}$ | Pearson | 0.377 |
| Sign. (two tails) |  | <0.001 |
| N |  | 118 |

Georeferencing of respondents to each building in the area was carried out by matching the area of residence and the construction date of the building in which the respondent resides. Each building was assigned the average value of the Index of Knowledge and Experience of risk and Inclination to Change of the coding pair (Residence and Construction date). From the attribution to each building of the values obtained from formulas (4) and (5), the distribution of the Risk Knowledge and Experience Index and Inclination to Change Index in the investigated area was developed in ArcGIS® support (Figure 7a,b, respectively). The maps make it possible to identify by individual building the mean values of the Risk Knowledge and Experience Index, within the range $2.61 \leq I_{KE} \leq 3.30$ and the mean values of the Inclination to Change Index, within the range $1.94 \leq I_{IC} \leq 2.54$. Perceived risk assessment was then developed according to formula (6). The exponential weights of the formula were kept constant and equivalent for both indices.

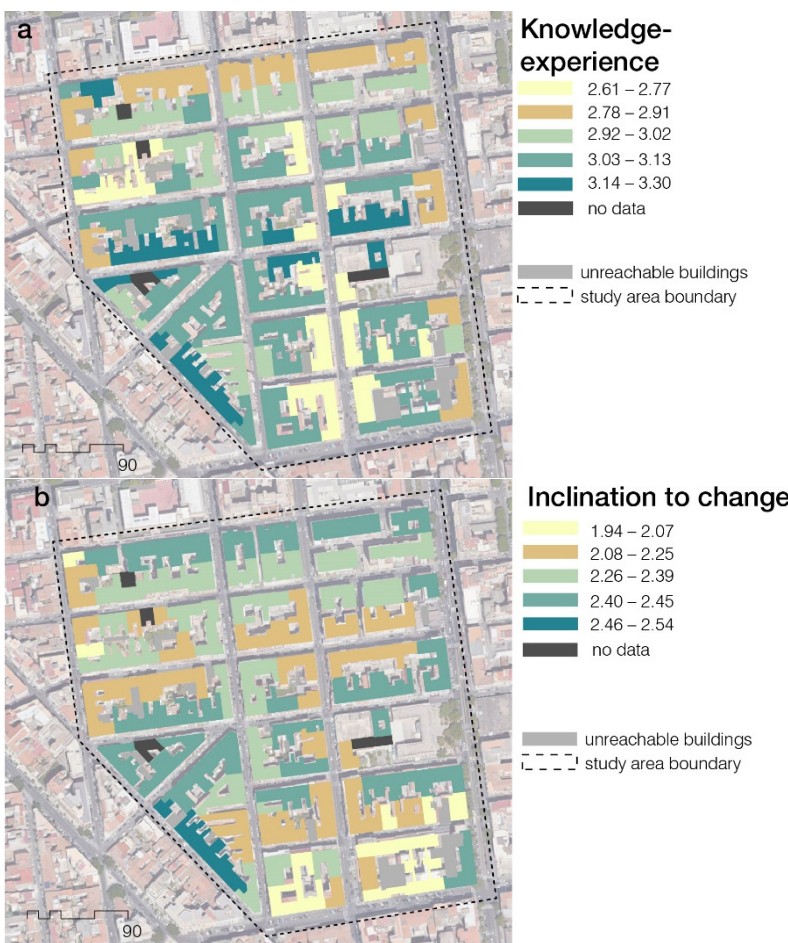

**Figure 7.** Spatial distribution of the two indicators and attribution to buildings in the investigated area: (**a**) Index of Knowledge and Experience of risk; (**b**) Index of Inclination to change.

## 4. Results and Discussion

In Figure 8, the distributions of the values of both the objective risk and the perceived risk in the considered district, calculated as explained in the previous section, are reported by means of color intensity scales.

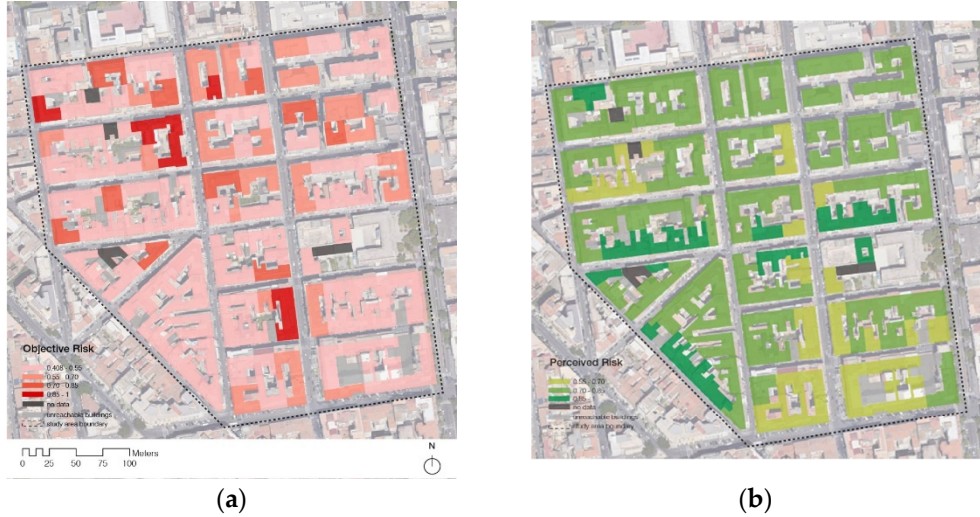

(**a**)          (**b**)

**Figure 8.** Spatialization of: (**a**) Objective risk and (**b**) Perceived risk of the investigated area.

The spatial distribution of the objective risk in the investigated area is shown in Figure 8a. The risk mapping, developed for each building unit, shows that for none of the 164 analyzed buildings the risk is zero: in fact, the range of variability in objective risk is between 0.41 and 1.00. The spatial distribution of perceived risk in the investigated area is shown in Figure 8b. In this case, the resulting perceived risk show values between 0.70 and 0.85.

The same values of objective risk and perceived risk calculated for each building in the investigated area are reported as points in the graph shown in Figure 9, with $R_o$ in abscissa and $R_p$ in ordinate.

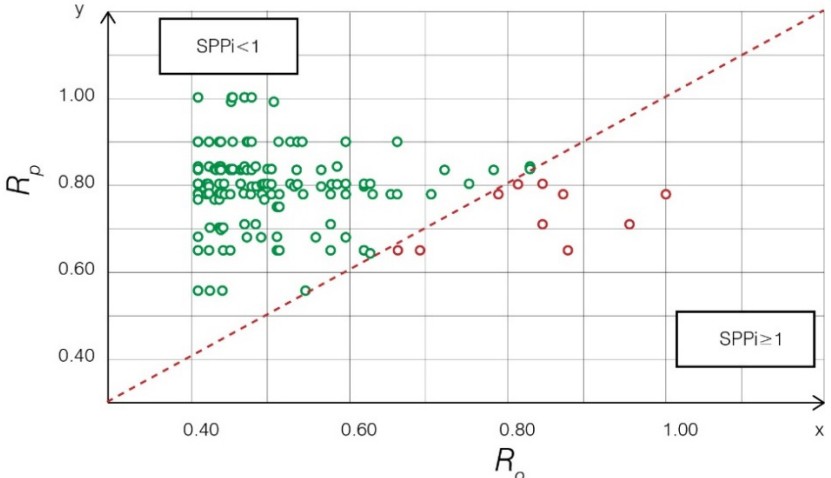

**Figure 9.** Risk comparison. The bisector marked in dashed red represents the locus of points where the ratio of objective risk to perceived risk is one. Red points represent the region where the objective risk is greater than perceived risk and for which Policies P1 apply. Green points represent the region where the objective risk is lower than perceived risk and for which Policy P2 apply.

Two main sectors can be identified in the graph, according with the value of the corresponding Seismic Policy Prevention Index (SPPi)$_j$ calculated through the ratio between with $R_o$ and $R_p$ (see Equation (5)):

**SPPi $\geq$ 1 sector**: In this region, the objective risk is greater than the perceived risk. The policy index values are between 1 and 1.352 (red dots below the bisector).

**SPPi < 1 sector:** Objective risk is lower than the perceived risk. The values of the policy index are between 0.408 and 1 (green dots above the bisector).

These sectors help us to identify the types of policies calibrated to the specific objective and perceived risk conditions designed to guide the way fiscal incentives are administered for seismic risk mitigation and adaptation in the investigated areas, as well as policies for expanding risk perception. Two possible policies that could be adopted for each area are outlined below.

### Sector I (SPPi $\geq$ 1)—Policy P1

In Sector I, the Seismic Policy Prevention Index is greater than one. This represents the most critical situation because perception underestimates the real risk condition of the building, implying a public disregard for seismic risk in the face of a real hazardous condition. Adopting measures to reduce objective risk appears to be the dominant policy, along with increasing risk perception.

Interventions can be twofold: on the one hand, the reduction in seismic vulnerability characterized by structural actions on the building unit; on the other hand, mechanisms to broaden risk perception through structural policies of public awareness. Since the objective risk assessment model is based on the components of exposure, vulnerability, and hazard, possible interventions related to the reduction in exposure and socio-demographic vulnerability are marginal (except in the case of buildings with high crowding due to their particular functionalities, for which the downgrading of the building is envisaged, with

the transfer of functions to safer facilities). Instead, hazard reduction through a systematic action of securing and seismically upgrading existing buildings can be an effective strategy for risk reduction. Vulnerability reduction can be pursued through specific interventions, aimed at removing critical issues (elements of vulnerability) of the building if present and/or by increasing the overall capacity of the structure having set the objectives to be achieved, in terms of seismic performance. The minimum level of safety required of a building is established by the current Technical Standards for Construction (NTC) 2018. All constructions built in the territory confront this standard, and existing buildings may have a safety level equal to that required by the technical standards for new buildings, or usually lower, with reductions often proportional to the age of the building itself. Assuming the minimum safety level of a building designed with modern regulatory standards to be 1, the existing building will have a safety value between 0 and 1.

In the area where the perceived risk is low, for certain risk mitigation interventions not to remain in the sphere of public intervention alone, it is necessary to stimulate the activation of participatory processes. For residents to be inclined to activate incentive mechanisms, forms of defiscalization can be provided.

The procedures for activating these operations consist of:

- Direct public interventions;
- Complex recovery actions;
- Public–private rewards (change in urban use, transfer of cubature).

The instruments described can make use of the principle of tax collection. For example, the reform of the building cadastre allows the transformation of the parameters for defining property values that form the tax base for local property and business taxes. Urban cadastral microzonation defines the value of properties area by area, consequently, defining the amount of taxes to be paid. Thus, an increase in taxes can be expected for those buildings that have not carried out maintenance work for the purpose of seismic retrofitting and improvement, a value that also drags on when the property is bought or sold as it is certified by a certificate of seismic non-compliance. To this end, it is possible to think about modulations in the application of the tax, either by introducing differentiated rates or by thinking about some tax deductibility, in order to create the necessary incentives for the implementation of seismic risk reduction policies. For building units where the perception of risk is low, in order to activate the mechanisms described above, in addition to incentives, it is necessary to raise awareness among the resident population for participation in routine and extraordinary maintenance processes and to avoid the dispersion of incentives on unnecessary interventions. The positive correlation between knowledge and experience of risk and propensity to change shows that the greater the perceived risk the greater the propensity to adopt risk mitigation measures, facilitating voluntary "adjustment" processes. The possibility of activating these mechanisms needs a "formative'/'informational" phase of actors (residents), through processes of public participation in "risk governance" that aims at increasing traditionally understood levels of knowledge and activating a process of fertilization between different actors engaged in the adoption of safety policies. Information and communication are posited as complementary mechanisms in which scientific knowledge and "non-standard knowledge" mix to produce a wider range of perspectives. Participation, thus, has steering power by creating alternative representations, leading to the development of problem perceptions, adding problematicity to the issue of risk, structuring problems collectively and interactively. To this end, in public policies for risk prevention, the methods of participation become necessary, deduced from experiences gained in North American countries as early as the 1960s [52–56] and in Northern Europe in the 1980s and 1990s [57–61], which, by facilitating public debate, contributed to the decision-making process. The policymaker, through information campaigns and periodic focus groups, facilitates and nurtures participatory processes [62], providing periodic opportunities for collective discussion and organizing training initiatives aimed at mutual learning. The intervention of the policy maker has the role of connecting the actors involved in urban redevelopment for the purpose of earthquake prevention in the most

exposed areas, expanding the boundaries of risk knowledge from the preventive phase to the emergency management phase, and to the post-emergency phase.

**Sector II (SPPi < 1)—Policy P2**

In Sector II, the Seismic Policy Prevention Index is less than 1. However, there are areas where the value is less than one for high values of both objective and perceived risk, and areas where the ratio is less than one for low values of both perceived and objective risk. When the perceived risk is high, the positive correlation between perceived risk and propensity to change leads to pro-activity with respect to the inclination to take risk mitigation measures and also facilitates voluntary mitigation processes by individuals. This is the ideal situation, especially to the extent that there is no zero-objective risk, though it can be low in relation to perceived risk.

One of the biggest problems in the implementation of rehabilitation interventions aimed at seismic risk reduction stems from the presence of a highly fractionated ownership structure that results in difficulties of action. In this sense, the way forward is the encouragement of interventions in the common parts of buildings. In addition, the use and promotion of forms of consortia and unitary interventions can be useful in dealing with the issue at the building level: the entities that individually or united are able to mount the operation with greater content of collective utility (in terms of works or services) are "rewarded" by a higher public economic contribution than the incentives traditionally provided. In this case, the involvement of condominiums understood as the minimum unit of analysis is expected. Policy P2 allows access to higher deductions when interventions are carried out on common parts of condominium buildings, aimed at both seismic risk reduction and other forms of complementary upgrading, such as energy upgrading. Buildings falling in this quadrant will be characterized by upgrading to one or more lower seismic classes through participation in the mechanism in condominium form.

Policies P2, due to the high values of perceived risk, aim to privilege, through appropriate mechanisms, those interventions operated by multiple actors, on the building stock that start from structural assessments rather than from individual building units, considering, in particular, structural connections and interrelationships between the different parts of a building or building aggregate. Urban planning or tax policies and mechanisms could create conditions for obtaining, from each individual intervention, greater collective benefits.

Agreements that could be experimented, as the possibility of favoring the transfer of cubature, if a building unit not subjected to adequate rehabilitation interventions is recognized as dangerous for the aggregate or the part of the settlement within which it is located; if this proves useful or necessary for risk reduction at the urban scale. This issue should be reconsidered within a "strategic" approach that definitively overcomes the "windfall" approach typical of the current administration of incentives.

Cubature transfer could be incorporated into the plan mechanism that strategically identifies cubage landing zones according to the logic of Transfer Development Rights (TDR) [63].

The graph in Figure 10 summarizes what has been said by plotting the value of the Seismic Policy Prevention index as a function of decreasing levels of objective risk. More precisely, on the *x*-axis is reported the code of the various building units arranged in descending order of objective risk.

The urgency of policy intervention in the urban area is hierarchized according to the vulnerability characteristics of the assets, expressed by the objective risk. The policies to be adopted then depend on the comparison with the corresponding perceived risk. To the building units with the highest values of objective risk, in the left part of the graph and above the value of SPPi = 1, policies P1 are likely to be applied (red dashed line sector); on the contrary, for the units below the value of SPPi = 1, policies P2 are recommended (green dashed line sector), at least above the average value of objective risk Ro (blue vertical line). Below this average value (yellow dashed line sector), the necessity of intervention is obviously less stringent (we could refer to this situation as a third policy P3).

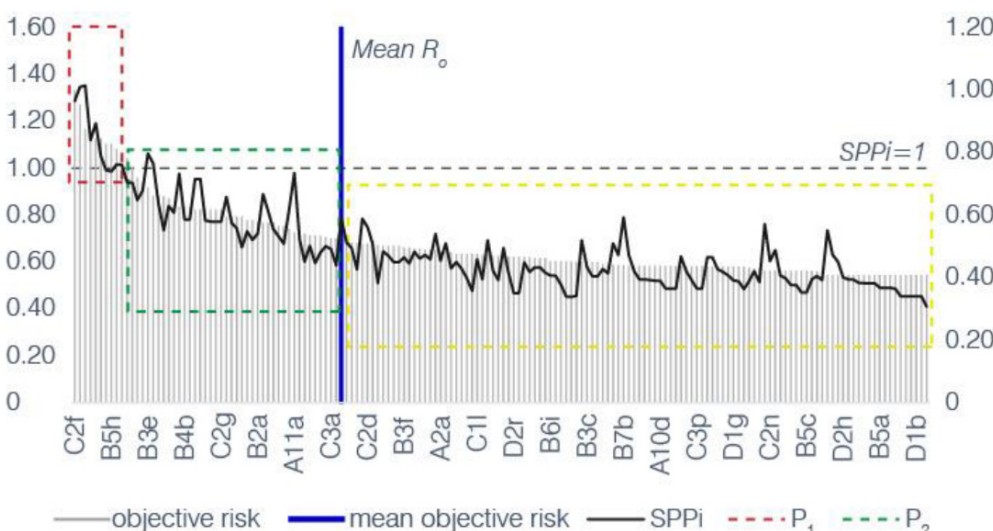

**Figure 10.** Seismic policy prevention index, SPPi versus objective risk.

Finally, the spatial distribution of the Seismic Policy Prevention index (SPPi)$_j$ in the considered neighborhood is shown in Figure 11, organizing the values for the various buildings inside three color classes according to the corresponding policy we suggest adopting.

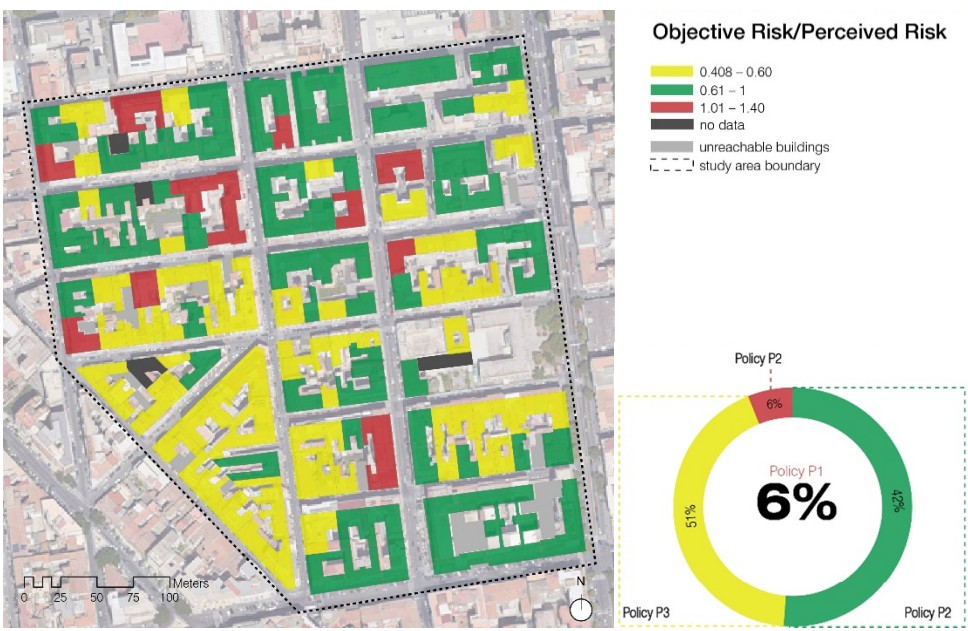

**Figure 11.** Spatial distribution of the Seismic Policy Prevention Index (SPPi)$_j$ in the considered district.

The percentage of buildings in a critical situation, thus requiring policy P1, appears to be quite low, about 6%, indicating that the district chosen as the case study is, after all, not particularly worrying for the policymaker. The latter feature emerges also from the calculation of the average value (SPPi)TA that can be also performed for the considered urban transformation area according to Equation (6). Such a value can been reported, by a color scale of increasing intensity, in the larger map shown in Figure 12, together with the analogous values of (SPPi)TA calculated for other districts within Catania's urban area. This kind of information should help the policy maker to plan targeted interventions based on the real needs of the administrated area.

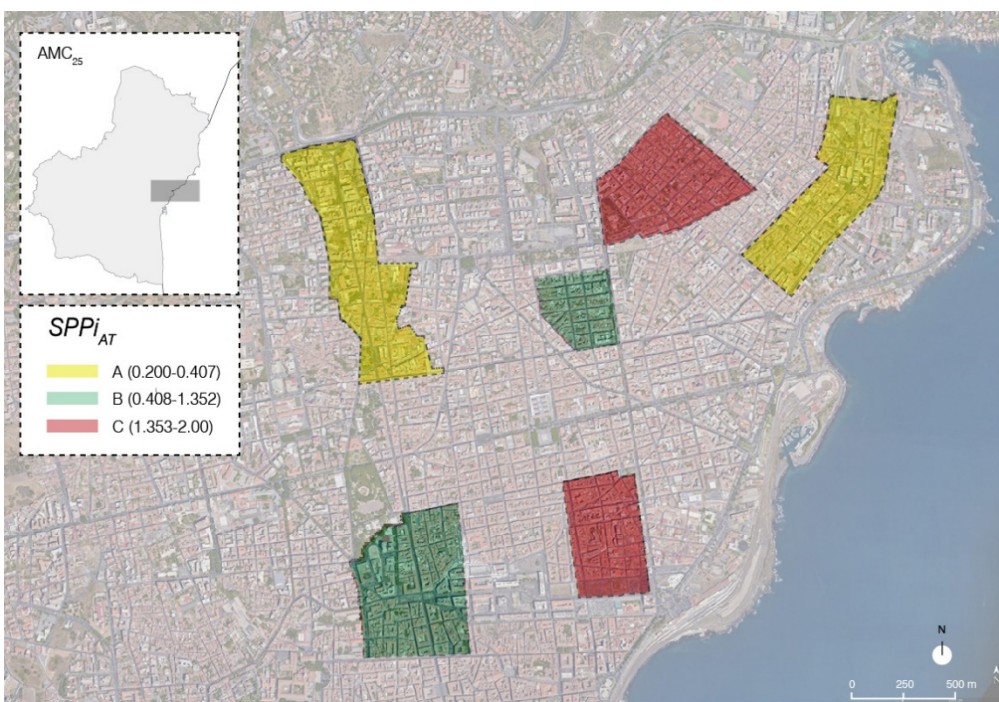

**Figure 12.** Map of the Seismic Policy Prevention Index (SPPi) for several districts in the urban context of Catania, showing the three categories, A-B-C. The central one is that reported in Figure 8.

## 5. Conclusions

The methodology proposed in this paper, applied to the case study of a portion of the urban area of Catania (Italy), makes it possible to identify risk levels for both individual buildings and entire neighborhoods through the introduction of a Seismic Policy Prevention index (SPPi). Intervention priorities are classified in relation to the areas in which the perceived risk is lower than the objective risk: this allows the adoption of specific policies oriented according to a criterion of risk perception enlargement.

This methodology is a tool that allows the identification of areas of conflict between public perception of risk and objective/physical risk, attempting their resolution through the identification of policies to be implemented to reduce this gap and encourage the implementation of earthquake prevention interventions. To this end, the perception of risk can be expanded through the definition of a network of districts in which permanent observatories of territorial participation are established as an operational tool for interaction between policymakers and public and private actors.

On the basis of our findings, it is easy to imagine future developments of this research line. For example, through an agent-based model, it could be possible to explore the policies introduced and assess their acceptability. In fact, one of the most complicated aspects in planning policies to be adopted in urban planning lies in their acceptability. Identified policies can simultaneously become a site of confrontation between stakeholders, public and private parties, due to their unworkability. An agent-based model would allow an ex-ante evaluation of proposed policies by understanding in advance the potential impact of the alternatives introduced, in order, on one hand, to support a participatory process of the actors involved, and on the other hand, to predict the reaction of the stakeholders to the new proposals and the expected results by means of a process of actor interaction, aiming at consensus building. The output of simulations can be used to understand what conditions are favorable for the convergence of opinions among stakeholders and which policies should be considered as better candidates because they are more likely to be accepted by stakeholders. An evaluation of the simulated effects of policies could then be adopted to correct and refine the inserted policies, so as to reach a scenario that satisfies most of the stakeholders involved.

**Author Contributions:** Conceptualization, E.F., A.G. and A.P.; methodology, A.G., A.P. and A.E.B.; software, E.F. and A.P.; formal analysis, A.G., A.P. and E.F.; investigation, E.F.; resources, E.F.; data curation, E.F. and A.P.; writing—original draft preparation, E.F.; writing—A.G., A.P., F.M., A.E.B. and A.R. All authors have read and agreed to the published version of the manuscript.

**Funding:** This research was partially funded by the University of Catania, with the projects "Linea di intervento 2 Piano PIA.CE.RI 2020/2022" of the Departments of Civil Engineering and Architecture and Physics and Astronomy "Ettore Majorana" and by the Italian Ministry of University and Research (MUR) with the projects "PRIN2017 linea Sud: Stochastic forecasting in complex systems" and PRIN2020 #20209F3A37.

**Informed Consent Statement:** Informed consent was obtained from all subjects involved in the study.

**Data Availability Statement:** Not applicable.

**Conflicts of Interest:** The authors declare no conflict of interest.

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
