# Peer review of "Objective and Perceived Risk in Seismic Vulnerability Assessment at an Urban Scale"

_sustainability, doi:10.3390/su14159380_

Round 1

Reviewer 1 Report

The paper is novel and interesting. It is well written and theoretically correct. Results are monitored and discussed well. This article can be a valuable addition to the available technical literature in this field. However, some minor comments should be considered for further improvement of the study.

1. Many studies have been published after the Amatrice earthquake on August 24th, 2016 to determine useful seismic vulnerability assessment tools at the urban scale. The introduction section must be completely rewritten to review and discuss such studies to show the difference and advantages of the proposed methodology.

2. The articles published in the previously published special issue in the journal "Seismic Vulnerability Assessment at Urban Scale" should be reviewed, discussed, and cited.

3. All the figures should be replaced with new ones to show a better resolution. 

Author Response

The paper is novel and interesting. It is well written and theoretically correct. Results are monitored and discussed well. This article can be a valuable addition to the available technical literature in this field. However, some minor comments should be considered for further improvement of the study.

R: We wish to thank the reviewer for the positive comments on the paper and for the precious suggestions that we tried to address as better described hereafter.

  1. Many studies have been published after the Amatrice earthquake on August 24th, 2016 to determine useful seismic vulnerability assessment tools at the urban scale. The introduction section must be completely rewritten to review and discuss such studies to show the difference and advantages of the proposed methodology.
  2. The articles published in the previously published special issue in the journal "Seismic Vulnerability Assessment at Urban Scale" should be reviewed, discussed, and cited.

R: The introduction has been modified according to the reviewer’s comments. We have included and commented on several recent papers on simplified seismic vulnerability assessment procedures and risk perception. The latter topic has been more deeply analysed since the relationship between risk assessment and risk perception represents the main focus of the paper

  1. All the figures should be replaced with new ones to show a better resolution.

R: As recommended, we replaced all images that had low resolution

Reviewer 2 Report

The manuscript deals with the an objective and perceived risk in seismic vulnerability assessment at urban scale.

The content of the paper corresponds to the topic stated in the title.

The article has a well thought-out, logical structure. It has been correctly divided into five chapters:

  1. Introduction
  2. Conceptual Framework and Methodology
  3. Material and Method
  4. Results and Discussion
  5. Conclusions

However, after chapter 4, chapter 5 is missing and there is a chapter 6 straight on.

The work contains 12 figures and 6 tables.

All diagrams should include units on the x-axis and y-axis. This should be added.

The aim of the study is to calibrate opportune policies, which allow addressing the most appropriate seismic risk mitigation options with reference to current levels of perceived risk.

The aim of this study has been achieved.

The selection of sources is correct. Sources include 46 references. However, the bibliography is from 1959 to 2018, and no item was cited after 2018.

The text contains all the reference (from [1] to [46]).

Author Response

The manuscript deals with the objective and perceived risk in seismic vulnerability assessment at urban scale.

R: We wish to thank the reviewer for the positive comments on the paper and for the precious suggestions that we tried to address as better described hereafter.

The content of the paper corresponds to the topic stated in the title.

The article has a well thought-out, logical structure. It has been correctly divided into five chapters:

  1. Introduction
  2. Conceptual Framework and Methodology
  3. Material and Method
  4. Results and Discussion
  5. Conclusions

However, after chapter 4, chapter 5 is missing and there is a chapter 6 straight on.

R: We corrected the numeration of the chapters.

The work contains 12 figures and 6 tables.

All diagrams should include units on the x-axis and y-axis. This should be added.

R: As suggested, we added the x-axis and y-axis in all diagrams.

The aim of the study is to calibrate opportune policies, which allow addressing the most appropriate seismic risk mitigation options with reference to current levels of perceived risk.

The aim of this study has been achieved.

The selection of sources is correct. Sources include 46 references. However, the bibliography is from 1959 to 2018, and no item was cited after 2018.

R: We have added some references regarding more recent literature on the topic of risk assessment and risk perception, mainly in the Introduction chapter.

The text contains all the reference (from [1] to [46]).

Reviewer 3 Report

The paper is well written, structured, and illustrated. It deals with a current theme that deserves all the attention. The paper determined the Seismic Prevention Policy Index (SPPi) and applied it to a region of Catania, which made it possible to identify risk levels for both individual buildings and entire neighborhoods, with a view to planning policies to be correctly adopted in urban planning, for seismic risk mitigation.

It should be made explicit that the subjective nature of the perception of risk by the different actors can influence the results.

The results were obtained and the conclusions are consistent.

The paper is well supported by the bibliography, but some of it is old. It is not possible to obtain more recent bibliographic sources, especially with regard to Risk Assessment and Perception of Risk?

minor corrections

Fig. 2 – no inhabitants>74< 6 years old/building

Line 216-217 – the resistance of construction materials decreases, not only because of corrosion phenomena. It is suggested that you indicate other main causes

Line 222 – it is also necessary to mention the "in situ" tests for the evaluation of structures, that do not require sample extraction.

Table 2 - s units?

Line 413 - give examples of safety characteristics that they are aware

Fig 9 – Correct SPPi <1

The text does not contain the reference [34]

Author Response

The paper is well written, structured, and illustrated. It deals with a current theme that deserves all the attention. The paper determined the Seismic Prevention Policy Index (SPPi) and applied it to a region of Catania, which made it possible to identify risk levels for both individual buildings and entire neighborhoods, with a view to planning policies to be correctly adopted in urban planning, for seismic risk mitigation.

R: We wish to thank the reviewer for the positive comments on the paper and for the precious suggestions that we tried to address as better described hereafter.

It should be made explicit that the subjective nature of the perception of risk by the different actors can influence the results.

R: Thank you, we stressed the above consideration in the introduction.

The results were obtained and the conclusions are consistent.

The paper is well supported by the bibliography, but some of it is old. It is not possible to obtain more recent bibliographic sources, especially with regard to Risk Assessment and Perception of Risk?

R: In the Introduction chapter, we added some more recent literature on the topic of risk perception and risk assessment: Marshall (2020); Saito et al. (2022); Quereshi et al. (2021).

minor corrections

Fig. 2 – no inhabitants>74< 6 years old/building

R: Corrected.

Line 216-217 – the resistance of construction materials decreases, not only because of corrosion phenomena. It is suggested that you indicate other main causes

R: We added other causes: «Infact it is well known that the resistance of construction materials decreases over time due to material quality, corrosion phenomena caused by atmospheric agents, stress state, rate and possible repetitiveness of load application».

Line 222 – it is also necessary to mention the "in situ" tests for the evaluation of structures that do not require sample extraction.

R: We have updated the sentence as follows: «A detailed evaluation of the seismic vulnerability of a building would therefore require an accurate level of knowledge of the structure and of its maintenance conditions and this can only be pursued following extensive structural investigations with ‘in situ’ tests».

Table 2 - s units?

R: Added

Line 413 - give examples of safety characteristics that they are aware

R: Examples added: «Descriptive analysis found that only 14% of respondents report having high or medium-high knowledge of the seismic safety characteristics of their property (namely building code, construction type) and the seismicity of their city of residence».

Fig 9 – Correct SPPi <1

R: Corrected

 The text does not contain the reference [34]

R: The reference is now at line 178.